# A 30 m resolution dataset of soil and water conservation terraces across China for 2000, 2010, and 2020

Enwei Zhang<sup>1</sup>, Yueli Chen<sup>3</sup>, Shengzhao Wei<sup>1</sup>, Chenli Liu<sup>1</sup>, Hongna Wang<sup>1</sup>, Bowen Deng<sup>1</sup>, Honghong Lin<sup>1</sup>, Xue Yang<sup>1</sup>, Yawen Li<sup>1</sup>, Xingwu Duan<sup>1,2,\*</sup>

- <sup>1</sup>Yunnan Key Laboratory of Soil Erosion Prevention and Green Development, Institute of International Rivers and Eco-Security, Yunnan University, Kunming, 650500, China
  <sup>2</sup>State Key Laboratory for Vegetation Structure, Function and Construction, Yunnan University, Kunming, 650500, China
  <sup>3</sup>State Key Laboratory of Severe Weather Meteorological Science and Technology, Chinese Academy of Meteorological Sciences, Beijing, 100081, China
- 10 Correspondence to: Xingwu Duan (xwduan@ynu.edu.cn)

Abstract. Terraces, as one of the most widely distributed and heavily invested soil and water conservation (SWC) measures in China, currently lack a comprehensive database containing spatiotemporal distribution and diverse classification types. This absence significantly hampers the accurate soil erosion assessment and SWC planning in China. To address this gap, we developed a two-stage mapping framework to classify various terrace measures and produced a new dataset named the Soil and Water Conservation Terrace Measures Dataset (SWCTMD). The dataset, spanning the years 2000 to 2020, was produced by integrating time-series Landsat imagery and digital elevation model data. The data incorporate SWC measure factors and four terrace types: level terraces, slope terraces, zig terraces, and slope-separated terraces. On average, the SWCTMD achieved OA of 91.7% and F1 of 83.3% for terraces, and 89.4% OA and 78.9% F1 for different terrace types, underscoring its high accuracy in terrace mapping. Comparative analysis demonstrated the superior robustness of the SWCTMD compared to existing products. This dataset demonstrated that terraces in China are predominantly concentrated in the Loess Plateau, Southwest and Southeast regions. From 2000 to 2020, the total terrace area increased by 41,594.1 km<sup>2</sup>, with slope terraces exhibiting the largest expansion, while decreases were primarily observed in peri-urban areas. Notably, the modeling results indicated that terraces had reduced soil erosion of cropland by approximately 1,390 million tons in 2020. The SWCTMD can be employed to enhance the accuracy of soil erosion simulations and support long-term analysis of soil erosion trends. Furthermore, the dataset provides valuable applications for earth system modelling and contributes to research on land resource management, food security, biodiversity, and water cycle. The SWCTMD is freely available at https://doi.org/10.11888/Terre.tpdc.302400 (Duan, 2025).

## 1 Introduction

Agricultural terraces are one of the most common cultivation techniques in mountainous and hilly areas, varying in shape and size. They consist of a flat cultivated section and nearly vertical risers. The risers are typically protected by dry stone, grass, scrub, or trees, and range from a few centimeters to several meters in height, with continuous or intermittent profiles

(Arnáez et al., 2015). Terraces form an important soil and water conservation (SWC) measure (Wickama et al., 2014; Londero et al., 2018). Based on the structures of the field surface, terraces can be categorized into level terraces, slope terraces, zig terraces, and slope-separated terraces (Liu et al., 2013a). By reshaping the surface microtopography, terraces decrease the slope length and gradient and alter hydrological pathways (Deng et al., 2021). These changes reduce soil erosion and runoff, improve water and soil conservation, and increase crop yields (Adgo et al., 2013; Chen et al., 2017, 2020; Wei et al., 2021). Within established soil erosion assessment frameworks, terraces have been incorporated as a support practice factor in the Universal Soil Loss Equation (USLE) and the Revised Universal Soil Loss Equation (RUSLE) (Wischmeier and Smith, 1978; Renard et al., 1997). In the Chinese Soil Loss Equation (CSLE), terraces are specifically represented as an SWC engineering practice factor (Liu et al., 2020). However, many large-scale assessments of soil erosion neglect this factor due to insufficient data on the spatial distribution of terraces (Gobin et al., 2004; Teng et al., 2016). Therefore, the mapping of terraces is crucial for soil erosion research.

Efforts have been made to map terraces in China. Three primary methods have been employed to obtain spatial extent and location information of terraces. The first method is a government-initiated land resource survey. Terraces were considered in paddy field surveys during the second and third nationwide land surveys in China. Terraces located in extensive drylands, particularly on steep slopes, were often categorized simply as dryland or irrigated land, without distinguishing terrace types. The second method is to extract terrace information from land use data (Liu et al., 2021). Existing land use products in China, such as FROM-GLC, GlobeLand30, CLCD, CACD, and GLC FCS30, generally classify terraces as cropland (Yu et al., 2013; Chen et al., 2015; Yang and Huang, 2021; Zhang et al., 2021; Tu et al., 2024). Among these, only the CNLUCC land use product further subdivides cropland into paddy field and dryland; however, this product also fails to distinguish terrace types on dryland (Liu et al., 2010). This limitation makes it challenging to extract information about terraces from existing land use data. The third method is to employ satellite images to identify terraces. For instance, Lu et al. (2023) employed deep learning methods to map terraces in the Loess Plateau based on high-resolution satellite images from October 2018 to February 2019. Li et al. (2024) produced a 30-meter resolution terrace map for China using 2017 Sentinel-2 imagery and Landsat-8 imagery on the Google Earth Engine (GEE) platform through the random forest (RF) algorithm. Similarly, Cao et al. (2021) produced a 30-meter resolution terrace map using 2018 Landsat-8 imagery and the RF algorithm on the GEE platform (Table 1). Although these maps have been widely used in soil erosion research (Li et al., 2023; Zhang et al., 2023), the limited classification of terrace types and the lack of long-term coverage restrict broader application at regional or national scales.

Table 1. Existing terrace products in China.


| Method        | Algorithm              | Study area Data               | Reference |  |  |
|---------------|------------------------|-------------------------------|-----------|--|--|
|               | Mapping terraces based | The Loess Google Earth images | Lu et al. |  |  |
| Deep learning | on the UNet++ deep     | Plateau                       | (2023)    |  |  |
|               | learning network       |                               |           |  |  |

| Machine learning | Mapping terraces based | China | Landsat-8 imagery and | Li et al.  |  |
|------------------|------------------------|-------|-----------------------|------------|--|
|                  | on the RF algorithm    |       | Sentinel-2 imagery    | (2024)     |  |
| Machine learning | Mapping terraces based | China | Landsat-8 imagery     | Cao et al. |  |
| Machine learning | on the RF algorithm    |       |                       | (2021)     |  |

The effectiveness of terraces in SWC varies according to type. Level terraces, characterized by flat cultivated surfaces, can effectively reduce the amount, velocity, and energy of surface runoff and increase water infiltration, thereby effectively preventing the transportation of sediment (Wei et al., 2012; Chen et al., 2013; Arnáez et al., 2015). Zig terraces increase water infiltration and reduce runoff by creating micro-catchments (Wang et al., 2004). Conversely, slope terraces, with their uneven surfaces, are more prone to generating runoff than level terraces or zig terraces (Wei et al., 2016). Level terraces exhibit the most effective SWC benefits (Oliveira et al., 2012). Compared to slope terraces, level terraces can reduce runoff by 56.5% and sediment by 53.1% (Chen et al., 2017). Ignoring terrace type can lead to inaccuracies in soil erosion assessment, and the absence of long-term terrace data hinders analyses of soil erosion trends.

Steep slope land accounts for more than one-third of the total cropland area in China. In recent decades, the construction of agricultural terraces has been the primary engineering measure for managing steep slope cropland (Liu et al., 2013b; Feng et al., 2017; Zhang et al., 2017). However, the existing terrace datasets lack detailed classification of terrace types and are limited to single-year data. These limitations have hindered soil erosion assessment, prediction, and SWC planning. To address this gap, we developed a two-stage mapping framework for terrace classification on the GEE platform. The first stage distinguishes terraces from non-terraces, while the second stage focuses on identifying different terrace types. Using this mapping framework, we developed the first long-term (2000 to 2020) national Soil and Water Conservation Terrace Measures Dataset (SWCTMD) of China. The dataset incorporates a detailed classification system. The accuracy of SWCTMD was evaluated using validation samples and compared with existing terrace maps. Additionally, the terrace dataset was used to identify spatial and temporal changes in terraces across China and to assess the SWC benefits provided by terraces.

## 2 Methodology


Figure 1 illustrates the framework of 30-meter resolution terrace mapping. The workflow includes sample collection, feature calculation, classification implementation, post-classification processing, and accuracy evaluation. Detailed information on each stage of the terrace mapping process is provided below.

Figure 1. The framework for mapping terrace.

# 2.1 The classification system and interpretation symbols

According to the findings of China's First National Census for Water (FNCW) (Liu et al., 2020), we identified the four major types of terraces: level terrace, slope terrace, zig terrace, and slope-separated terrace. The interpretation keys for the different terrace types included shape, size, texture, color, and location (Table 2).

**Table 2.** Image characteristics of different terrace types.

| Terrace types | Image characteristics                                                          | Real photo |  |  |
|---------------|--------------------------------------------------------------------------------|------------|--|--|
|               | Steep slope land transformed into a series of successively receding flat       |            |  |  |
| Level terrace | surfaces, with bunds constructed from soil or stones, ranging in width from 5  |            |  |  |
|               | to 40 m, looking like the steps of a staircase in remote sensing images. In    |            |  |  |
|               | contrast to slope terraces, level terraces are predominantly found in low and  |            |  |  |
|               | flat areas.                                                                    |            |  |  |
|               | Similar to level terraces, but with wider and more uneven surfaces, these      |            |  |  |
| Clara tarraga | terraces exhibit irregular shapes in remote sensing images. They are primarily |            |  |  |
| Slope terrace | used for dryland agriculture and are largely distributed the areas with slopes |            |  |  |
|               | greater than 5°.                                                               |            |  |  |

|                 | Steep slope land has been transformed into step-like terraces that are narrower  |                                       |  |  |
|-----------------|----------------------------------------------------------------------------------|---------------------------------------|--|--|
|                 | than level terraces. The surfaces of these terraces exhibit regular strip shapes |                                       |  |  |
| Zig terrace     | in remote sensing images. These terraces are primarily found in sloping          |                                       |  |  |
|                 | regions and are used for planting permanent crops such as tea.                   |                                       |  |  |
|                 | Each flat surface constructed on steep slope land retains a segment of the       |                                       |  |  |
| Slope-separated | original slope above, forming a composite structure that features a slope        |                                       |  |  |
| terrace         | between flat surfaces. These terraces are primarily used for rubber              | erraces are primarily used for rubber |  |  |
|                 | plantations.                                                                     |                                       |  |  |

#### 2.2 Data and preprocessing

In this study, we primarily used Landsat surface reflectance (SR) data, the Copernicus digital elevation model (DEM) data, and GlobeLand30. Detailed information about these datasets is provided in Table S1.

#### 95 2.2.1 Landsat SR data


The study used Landsat-5/8 SR data, with a spatial resolution of 30 m and a temporal resolution of 16 days. The data were accessible through the GEE platform. The Landsat SR data from the sensors had been atmospherically corrected by the United States Geological Survey (USGS) utilizing the LEDAPS algorithm (Masek et al., 2006). These data included Quality Assessment (QA) masks that indicated the usability of the pixel data, produced using the CFMASK algorithm (Zhu and Woodcock, 2012). We used QA bands to identify and remove clouds and cloud shadows in each Landsat SR image, and missing data after cloud removal were filled using images from the previous year. Due to the inconsistency in the wavelength of band among different Landsat sensors (Roy et al., 2016), we used only Landsat-8 SR imagery for the SWCTMD in 2020, and Landsat-5 SR imagery for the SWCTMD in 2000 and 2010.

#### 2.2.2 Copernicus DEM

Topographical features are essential characteristics that differentiate regular cropland and terrace, playing a crucial role in the identification of terraces. We used the Copernicus DEM data to calculate these topographical features. The Copernicus DEM is a Digital Surface Model with 30 m resolution, derived from radar satellite data acquired from 2010 to 2015 during the TanDEM-X mission. Compared to other DEM data (SRTM, ASTER GDEM, ALOS World 3D, and NASADEM), Copernicus DEM has the highest accuracy among open-source data (Guth and Geoffroy, 2021), exhibiting the greatest detail of terrain (Li et al., 2022a). The GEE platform provides access to the Copernicus DEM at 30 m resolution.

## 2.2.3 GlobeLand30





To improve the accuracy and efficiency of terrace identification, we used the union of cropland data from GlobeLand30 from 2000 to 2020 as the range for terrace identification. Then, we removed cropland with a slope of less than or equal to 2° (Ministry of Natural Resources of the People's Republic of China, 2019). GlobeLand30 is a widely used land use dataset with 30 m resolution that employs a pixel-object-knowledge classification method, effectively utilizing the advantages of various classification algorithms (Chen et al., 2015). The accuracy of cropland area and spatial location of GlobeLand30 is higher than the other four products (FROM-GLC, GlobCover, MODIS Collection 5, and MODIS Cropland) in China (Lu et al., 2016). The cropland from GlobeLand30 includes paddy fields, drylands, pastures, and permanent crop lands (e.g., tea and coffee plantations). Therefore, we adopted the cropland from GlobeLand30 as the range of terrace classification.

# 2.3 Feature space construction

Feature variables play a crucial role in the classification of remote sensing images. In this study, we constructed an input dataset comprising four aspects: spectrum, spectral indices, phenology, and topography. The six optical bands (red, green, blue, near-infrared, shortwave infrared 1, and shortwave infrared 2) from Landsat SR imagery for a specific year, along with the corresponding spectral indices (NDVI, MNDWI, NDBI, BSI, LSWI, and EVI), were composited into the 25th, 50th, and 75th percentiles utilizing the metrics-composite method. The percentiles effectively represent phenological information while simplifying time series information, thereby reducing annual time series noise and enhancing the accuracy of classification (Duan et al., 2024). In addition to the Landsat-based metrics, we incorporated seven frequently utilized topographic features: slope, aspect, slope of slope (SOS), relief (RF), slope shape (P), roughness (R), and elevation (Tang et al., 2016). The calculation method for feature variables is shown in Table S2. To eliminate multicollinearity among the feature variables, we removed highly correlated features based on two criteria: (a) a variance inflation factor (VIF) value for each feature less than 10, and (b) pairwise Pearson correlation coefficients are below 0.7 (Liao et al., 2021). Detailed information about the used features is provided in Table S3, Table S4, and Table S5.

#### 2.4 Collection of training samples

Samples are a critical component in supervised classification. We used manual visual interpretation methods to obtain samples from the years 2000, 2010, and 2020. To ensure that the collected samples were evenly distributed across the study area, we implemented a strategy of gathering samples by subregions. The study area was divided into 1,641 subregions. Utilizing high-resolution images from Google Earth Pro software, we collected at least five samples from each subregion (Fig. S1). Through this method, we collected a total of 103,374 samples. Specifically, 34,891 samples were obtained in 2000, 34,072 samples were obtained in 2010, and 34,411 samples were obtained in 2020 (Table S6).

## 140 2.5 Ground-truth reference data




The terrace validation data were derived from the FNCW conducted between 2010 and 2012. These data were obtained through field surveys and provide detailed information about terraces, including terrace types and GPS coordinates. The survey covered croplands nationwide. A total of 14,986 survey sites were used for validation of terrace accuracy in 2010, comprising 3,706 terrace samples and 11,280 non-terrace samples (Fig. 2). The statistics for different terrace types are listed in Table S7. Based on these data, the terrace validation samples for 2000 and 2020 were obtained by overlaying high-resolution remote sensing imagery from Google Earth Pro for verification.

Figure 2. The Spatial distribution of validation samples.

#### 2.6 Terrace classification on the GEE platform

The GEE platform offers a variety of classification algorithms. We selected the widely used RF model for terrace classification, as the algorithm offers the advantages of remarkable performance, high efficiency, and interpretability (Rodriguez-Galiano et al., 2012; Gong et al., 2019). Two essential parameters must be set for the RF model. In this study, we set the number of trees to 500 and determined the number of variables per split as the rounded square root of the feature number. Other parameters were maintained at the default settings specified by the GEE platform (He et al., 2017; Gong et al., 2020). To alleviate the impact of crop spectral variability on classification accuracy, the study area was divided into six subregions (Fig. 3) (Li et al., 2024). The different terrace types within each region were classified separately. Given the sensitivity of the RF model to the ratio of samples across different classes (Chen et al., 2024), we implemented a two-stage

mapping approach for classifying terraces within each region. In the first stage, RF was utilized to differentiate between terrace and non-terrace classes. In the second stage, RF was utilized to classify various terrace types, including level terraces, slope terraces, zig terraces, and slope-separated terraces. In Stage I of the mapping process, samples from both terrace and non-terrace samples were used, whereas only terrace samples were utilized in Stage II.

**Figure 3.** Geographical regionalization in China. SW represents Southwest China. NW represents Northwest China. NENC represents Northeast and North China. SC represents South China. CC represents Central China. EC represents East China.

#### 2.7 Post-classification processing





Both supervised and unsupervised classification methods in remote sensing rely on the spectral characteristics of image pixels. A critical issue is the presence of isolated pixels in the classification results, which exhibit high local spatial heterogeneity between neighboring pixels (Hirayama et al., 2019). This phenomenon, commonly known as the salt-and-pepper effect, is regarded as noise affecting accuracy. Terraces, being primarily constructed in hilly or mountainous regions, often exhibit a scattered and irregular distribution, which leads to an obvious salt-and-pepper effect in classified images. Given the small areas of terraces, we applied a mode filter with 3 × 3 px for spatial filtering processing to mitigate the salt-and-pepper effect from the classification results. To improve the overall quality of the mapping results, we conducted spatial-temporal consistency checks to suppress illogical land use conversions. Specifically, for areas that were cropland in both the previous year and the current year (excluding grain-for-green areas), we modified those areas that were previously terraces but were identified as non-terraces in the current year to terraces.

## 2.8 Accuracy assessment

It is an essential step to assess the accuracy of the products prior to utilizing data in related applications. The classification maps were evaluated using a confusion matrix calculated from validation samples. The confusion matrix is widely regarded as the standard method for evaluating the accuracy of classified images. This method offers quantitative assessment metrics, including the kappa coefficient (KA), overall accuracy (OA), the producer's accuracy (PA), and the user's accuracy (UA), which collectively assess the performance of the products. OA and KA measure the total map accuracy. PA and UA measure the omission and commission errors for each class. In addition, we calculated the F1 score, which reflects the balance between UA and PA. The KA, OA, PA, UA, and F1 metrics range from 0 to 1, where 1 indicates optimal performance and 0 represents the poorest performance. The formula for the F1 metric is shown in Eq. (1):

$$F1 = 2 \frac{PA \times UA}{(PA + UA)}$$
 (1)

In this study, we constructed two confusion matrices: one to evaluate the accuracy of terraces and non-terraces, and the other to assess the accuracy of various terrace types.

#### 3 Results


#### 3.1 Overall accuracy assessment

Two confusion matrices corresponding to different terrace classification levels were generated using the validation samples. For the classification of terrace and non-terrace, the OA ranged from 91.7% to 91.8%, with KA ranging from 77.7% to 78.2%, and F1-scores ranging from 83.1% to 94.6% (Table 3). For terrace class, the UA ranged from 77.6% to 84.6%, and the PA ranged from 81.7% to 90.7%, and the F1 above 80%, indicating that the overall classification performs well.

OA (%) KA (%) Year **Types** UA (%) PA (%) F1 score (%) 97 94.5 Non-terrace 92.1 2000 91.7 78.2 Terrace 77.6 90.7 83.6 95.1 Non-terrace 94.1 94.6 91.8 77.7 2010 84.6 Terrace 81.7 83.1 Non-terrace 96.8 92.2 94.5 2020 91.7 77.8 Terrace 77.7 89.8 83.3

**Table 3.** The accuracy matrix for the terrace and non-terrace.


For different terrace types, the OA ranged from 88.8% to 89.8%, KA ranged from 65.1% to 69.5%, and F1 scores ranged from 68.9% to 93.9% (Table 4). Level terraces exhibited the highest classification accuracy, followed by slope-separated

terraces, slope terraces, and zig terraces. From the UA and PA, the commission errors were lower than the omission errors for different types of terraces. Level terraces had the lowest misclassification error among the terrace types.

**Table 4.** The accuracy matrix for the different types of terraces.




| Year | types                   | UA (%) | PA (%) | F1 score (%) | OA (%) | KA (%) |
|------|-------------------------|--------|--------|--------------|--------|--------|
| 2000 | Level terrace           | 93.7   | 94.1   | 93.9         | 89.7   | 66     |
|      | Slope terrace           | 70.1   | 70.6   | 70.3         |        |        |
|      | Zig terrace             | 74.6   | 64.1   | 68.9         |        |        |
|      | Slope-separated terrace | 85.7   | 70.6   | 77.4         |        |        |
| 2010 | Level terrace           | 93.8   | 94     | 93.9         | 89.8   | 69.5   |
|      | Slope terrace           | 73.1   | 73.2   | 73.2         |        |        |
|      | Zig terrace             | 77.6   | 68.6   | 72.8         |        |        |
|      | Slope-separated terrace | 83.3   | 88.2   | 85.7         |        |        |
| 2020 | Level terrace           | 93.7   | 92.9   | 93.3         | 88.8   | 65.1   |
|      | Slope terrace           | 67.8   | 71.3   | 69.5         |        |        |
|      | Zig terrace             | 70     | 68.8   | 69.4         |        |        |
|      | Slope-separated terrace | 86.7   | 72.2   | 78.8         |        |        |

Figure 4 illustrates the spatial consistency between the SWCTMD and two existing datasets: the 2018 China Terrace Map (CTM2018) (Cao et al., 2021) and the 2017 China Terrace Map (CTM2017) (Li et al., 2024). SWCTMD exhibited the highest accuracy. Compared to SWCTMD and CTM2018, CTM2017 exhibited relatively lower accuracy for both typical terrace and non-terraces areas (regions B, C, D F and G in Fig. 4b). For typical terraces, SWCTMD and CTM2018 show similar identification performance (regions A, B, C, D and F in Fig. 4b). However, for atypical terraces, such as zig terraces located in Yunnan Province, SWCTM successfully identified these as terraces, whereas CTM2018 failed to identify them as terraces (regions E in Fig. 4b). Conversely, for non-terrace areas situated in the Middle-Lower Yangtze River, SWCTMD accurately classified these as non-terraces, while CTM2018 erroneously classified them as terrace areas (regions G in Fig. 4b). At the provincial scale, the majority of provinces exhibit larger terrace areas in SWCTMD compared to both CTM2018 and CTM2017 (Tables S8 and S9).

**Figure 4.** Regional comparisons of the three terraces datasets. (a) The distribution of cropland in China in 2020. (b) The spatial distributions of the three terraces datasets.

## 215 3.2 Accuracy assessment in different regions


The classification of terraces across different regions performed well, but there were significant differences in accuracy among the regions. The Southwest and Northwest had the highest concentrations of terraces. Southwest China achieved superior classification performance due to its pronounced terrace morphology and spectral characteristics. Southwest China demonstrated the highest classification precision, with average values of UA, PA, F1, OA, and KA at 89.8%, 95.8%, 92.7%, 90.2%, and 77.9%, respectively (Table S10). Northwest China followed closely, with corresponding average values of 75.2%, 91.6%, 82.3%, 89.6%, and 75.1%. In contrast, Northeast and North China, South China, Central China, and East China have relatively flat terrain, with terraces being similar to the surrounding cropland, resulting in relatively lower classification accuracy. The mean F1 scores were 70.6%, 77.5%, 81%, and 73.8%, respectively. The mean OA scores were 94.5%, 91.3%, 87.8%, and 91.7%, respectively, and the KA were around 70% (Table S10).

The overall classification accuracy for different terrace types across all regions was well. Northwest China, Northeast and North China, Central China, and South China had the highest classification accuracy, followed by Southwest China and East China. The average UA, PA, F1, OA, and KA values of Northwest China, Northeast and North China, Central China, and South China were 82.3%, 81.1%, 81.5%, 90.4%, 67.9%. The average UA, PA, F1, OA, and KA values for Southwest China

and East China were 76.5%, 77.9%, 77.1%, 90%, 64.4% (Table S11). Among all terrace types, level terraces had the highest classification accuracy across all regions, followed by slope-separated terraces, slope terraces, and zig terraces.

#### 3.3 Spatiotemporal variation of terraces in China




Terraces are primarily distributed across the hills, basins, and plateaus of China (Figs. 5a and S2). The Sichuan Basin exhibited the highest concentration of terraces, followed by the Yunnan-Guizhou Plateau and the Loess Plateau. Terraces are also extensively found in the hilly regions of central and southeastern China. Level terraces are distributed in the gentler slopes of hilly regions in China. Sloped terraces are most densely distributed across Yunnan and the Loess Plateau, with lesser occurrence in the hilly regions of central and southeastern China. Zig terraces are mostly distributed in Southwest China and Northwest China, while slope-separated terraces are mostly located in Southwest China and Central China (Figs. 5a and 5b). In terms of spatial changes, the increasing terraces are mainly distributed in the Yunnan-Guizhou Plateau, the Loess Plateau, the Sichuan Basin, and South China from 2000 to 2020 (Fig. 6a). These areas are severely affected by soil erosion. Yunnan and Guangxi are the provinces with the largest increase in terraces (Fig. 6b). The decreasing terraces are mainly distributed around urban areas from 2000 to 2020, where urban expansion has occupied some terrace areas.

**Figure 5.** The spatial patterns of different terrace types at the pixel and provincial. (a) The spatial distribution of different terraces in China in 2020. (b) The different terrace areas in provinces in 2020.

**Figure 6.** The spatial changes of the terrace at the pixel and provincial. (a) The spatial changes in terraces from 2000 to 2020. (b) Changes in the terrace areas across different provinces from 2000 to 2020.

The provinces with the largest terrace areas were Sichuan, Yunnan, Guizhou, Gansu, Shanxi, Hunan, Shaanxi, and Chongqing, while other provinces had relatively smaller terrace areas (Fig. 7a). Among these, Chongqing, Sichuan, Guizhou, and Yunnan exhibited the highest percentage of terraces, with over 80% of cropland converted to terraces (Fig. 7b). From 2000 to 2020, Yunnan, Guangxi, Shanxi, and Shaanxi experienced the most significant increases in terrace areas, increasing by 11,372.4 km² (13.1%), 5,192.4 km² (32.9%), 2,395 km² (6.1%), and 2,295 km² (6.2%), respectively (Fig. 7a). In terms of terrace types, the areas of level terraces, slope terraces, zig terraces and slope-separated terraces increased by 5,701.4 km² (1.3%), 29,876.3 km² (18.9%), 5,886.5 km² (31.4%), and 129.9 km² (24.9%), respectively, with the slope terrace having the largest increase (Figs. 7c, d, e and f). Overall, China's total terrace area expanded from 612,885.4 km² in 2000 to 654,479.5 km² in 2020, an increase of 6.8% (Fig. 7g).

**Figure 7.** The changes of terrace areas at provincial and types from 2000 to 2020. (a) The changes of terrace area in different provinces. (b) The proportion of terraces to cropland in different provinces. (c-f) The areas of level terrace, slope terrace, zig terrace, and slope-separated terrace. (g) The total terrace areas of China.

## 3.4 SWC measure factor and responses of soil erosion to terraces




The SWC measure factor (E) value for each terrace measure was given according to the FNCW and published literature (Table S12) (Duan et al., 2020; Liu et al., 2020). Using these parameters, we generated spatial distribution maps of E for the years 2000, 2010, and 2020 (Fig. S3). With these data, we utilized the CSLE to assess cropland soil erosion across China in 2020 (Notes S1, S2 and S3). Figure 8 illustrates the soil erosion modulus under the terrace scenario in 2020. The average soil erosion modulus for cropland was 8 t·ha<sup>-1</sup>·y<sup>-1</sup>, with a total eroded area of 842,685 km<sup>2</sup>. Compared to the scenario without terrace measures, the average soil erosion modulus of cropland decreased by 7 t·ha<sup>-1</sup>·y<sup>-1</sup> (46.5%), and the erosion area was reduced by 223,134.8 km<sup>2</sup> (20.9%) (Figs. S4a and b). Collectively, terrace measures reduced approximately 1,390 million tons of cropland soil erosion, accounting for 46.5% of the total erosion on croplands. Spatially, the reduction in erosion was primarily concentrated in the Loess Plateau, Sichuan Basin, and Yunnan-Guizhou Plateau. Ningxia, Gansu, Sichuan, Chongqing, Qinghai, Guizhou, Shanxi, and Yunnan exhibited the largest decreases, with reductions of about 65%–75%.

**Figure 8.** The effects of terraces on soil erosion in different provinces. (a) The soil erosion alleviated by terraces. (b) The percentage represents the amount of soil erosion alleviated by terraces as a proportion of the total soil erosion without terraces.

#### 275 4 Discussion



#### 4.1 Comprehensive and Reliability of SWCTMD

We compared the 2020 terrace area estimated by SWCTMD with those from CTM2018 and CTM2017. SWCTMD exhibited the largest terrace area compared to CTM2018 and CTM2017. The areal discrepancies can be attributed to the following reasons. First, CTM2017 and CTM2018 predominantly focused on the most typical level terraces, whereas our research encompasses a broader range of terrace types, including non-typical terraces such as slope terraces, zig terraces, and slope-separated terrace. Second, each dataset employed distinct cropland for terrace classification. SWCTMD utilized the union of cropland with slopes exceeding 2° from the 2000, 2010, and 2020 GlobeLand30 cropland data, whereas CTM2018 employed only the 2010 GlobeLand30 cropland data, and CTM2017 adopted FROM-GLC cropland data. Third, CTM2018 excluded isolated patches smaller than 9,000 m² from its classification scheme. However, since SWCTMD constrains its classification to cropland with slopes exceeding 2°, the identified terrace areas in Anhui, Fujian, Jiangxi, and Zhejiang provinces were smaller than those from CTM2018. In these provinces, CTM2018 included terraces with slopes below 2°, which is classified

as non-terraces according to the technical regulations of the third nationwide land survey. Overall, our dataset provides more comprehensive coverage for terraces and exhibits higher accuracy and robustness.

## 4.2 Spatial pattern of terraces





The Sichuan Basin, Loess Plateau, and the Yunnan-Guizhou Plateau are the three regions with the highest concentration of terraces in China. Other areas, characterized by relatively gentle slopes, have fewer terraces. In the hilly areas of the Sichuan Basin and the Yunnan-Guizhou Plateau, humans have constructed terraces through a long-term process of adapting to nature by reshaping mountainous landscapes (Zhang et al., 2008; Duan et al., 2020). This process has also fostered unique cultural and social practices associated with terraces (Zhan and Jin, 2015; Zhang et al., 2024). These regions face challenges such as limited cultivated land resources, steep slopes, and intense precipitation (Liu et al., 2014; Li et al., 2016; Wang and Dai, 2020). The construction of terraces not only produces additional cultivable land but also optimizes water resource utilization and reduces soil erosion (Wei et al., 2017).

In recent years, the Chinese Land Consolidation projects and the Well-Facilitated Farmland projects have prioritized slope-to-terrace conversion as the primary land consolidation strategy in mountainous regions (Tang et al., 2019). This initiative has significantly increased the terrace area in southwestern China. In the Loess Plateau, terraces are primarily constructed for SWC and ecological restoration. Natural factors such as fragmented mountainous terrain, loose soil, and intense rainfall, combined with human activities of deforestation, overgrazing, and cultivation on steep slope, have made the Loess Plateau one of China's most severely eroded regions (Wang et al., 2010; Liang et al., 2015). Over the past few decades, large-scale programs such as Grain-for-Green and terrace construction initiatives have been implemented to combat soil and water loss (Fu et al., 2017). Most terraces in the Loess Plateau are dryland terraces, predominantly located in Gansu, Ningxia, Shanxi, and Shaanxi Provinces. In northeast China, cropland have long slope lengths, and gentle slope degrees (Liu et al., 2020), resulting in fewer terraces being built. In contrast, in the hilly regions of central and southeastern China, terraces have also been constructed despite gentler slopes. Unlike the Sichuan Basin, Loess Plateau, and Yunnan-Guizhou Plateau, where terraces serve as a necessity for managing steep terrain, the primary motivation in these areas is to expand the amount of land for the cultivation of economic crops such as tea and fruit trees (Li et al., 2022b). However, in mountainous and hilly regions, urban expansion has occupied some formerly terraced areas.

## 4.3 Soil conservation of terraces

Due to the lack of large-scale terrace distribution data, many previous continental-scale soil erosion assessments have generally not considered the influence of terraces, such as Europe, Australia, and Africa (Gobin et al., 2004; Teng et al., 2016; Salhi et al., 2025). This has led to overestimation of cropland soil erosion. Wang et al. (2021b) estimated cropland erosion at 1,939.7×10<sup>6</sup> tons in 2015 without accounting for terraces. Conversely, our study indicated cropland erosion at 1,599.4×10<sup>6</sup> tons in 2020, closely aligning with the 2011 FNCW result of 1,640.0×10<sup>6</sup> tons. Regarding the erosion reduction effects of terraces, Li et al. (2024) mapped terrace in 2017 and found that terraces reduced cropland erosion by 950 million

tons. In contrast, our study estimates that terraces reduced cropland erosion by 1,390 million tons in 2020. The discrepancy between the two results from Li et al. (2024) failure to distinguish between terrace types, resulting in an underestimation of terrace benefits.

According to our estimate, soil erosion of the Loess Plateau accounted for only 12.6% of the total cropland erosion. Terraces in this region contributed to 17.4% of the total reduction of cropland soil erosion, demonstrating the benefits of terraces to SWC. In Northeast China, terraces are sparse, and cropland is characterized by long slope lengths and gentle slope degrees, with erosion accounting for 27.6% of the total cropland erosion. In Southwest China, erosion amount accounted for 23.4% of total cropland erosion. Northeast and Southwest China should be the key areas for future soil erosion protection efforts. In Hebei, Henan, and Shandong Provinces, extensive cultivation and high crop planting intensity contributed 16.3% of the total cropland erosion, which warrants attention. In this study, each terrace type was assigned a fixed E value to facilitate the estimation of large-scale soil erosion. However, this approach overlooks spatial heterogeneity in terrace structure, maintenance status, field management, climate, and topography. Future research should incorporate regional characteristics and adjust the E-value accordingly.

## 4.4 Limitations and prospects







The spatial heterogeneity of land types frequently leads to class imbalance in remote sensing classification, consequently diminishing classification accuracy for minority classes that occupy a smaller area (Xiao et al., 2024). The models tend to favor majority classes during training, reducing their ability to accurately identify minority classes (Chen et al., 2025). When the ratio of samples across different classes remains balanced, classification performance typically falls short of optimal accuracy thresholds (Deng et al., 2025). A common strategy to alleviate this negative effect is to divide the study area into multiple sub-regions for localized classification, thereby reducing the impact of sample imbalance on model accuracy (Zhang et al., 2020). In this study, we employed a partitioned two-stage RF approach to reduce the effects of sample imbalance on classification accuracy. The results demonstrated that classification for terrace and different terrace types achieved satisfactory accuracy in both the entire study area and individual subregions. However, the accuracy metrics of the majority class were still higher than those of the minority class. In future studies, sample optimization techniques and more advanced classification methods could be combined to further improve the accuracy of minority class classification.

The complex and diversity diverse landform types have resulted in differences in the spectral information and topographic features of terraces in different regions. In Southwest and Northwest China, terraces exhibit concentrated distributions with clearly defined characteristics, making them easily identifiable. However, South China, Central China, and East China have relatively low topographic relief. Some terraces have spectral and topographic features similar to those of sloping farmland and flatland. This similarity, combined with the presence of mixed pixels in medium-resolution imagery (Wang et al., 2021a), makes it challenging to detect the terrace patches. Although the classification used 30-meter Landsat imagery in this study was generally robust, some fragmented and narrow terraces were omitted. Future research could employ high-resolution remote sensing images to effectively identify fragmented and narrow terraces. Previous small-scale studies have

demonstrated that the use of high-resolution remote sensing imagery, combined with object-based classification methods and deep learning approaches, can significantly enhance classification accuracy and reduce the impact of spectral confusion and mixed pixels on terrace identification (Diaz-Varela et al., 2014; Wang et al., 2023; Kan et al., 2025). To improve classification accuracy and efficiency, cropland data were used as the basis for terrace identification. Inevitably, the accuracy of cropland data impacts the terrace mapping process, as errors in cropland data propagate into the terrace maps. In summary, future studies could utilize high-resolution remote sensing imagery and more accurate cropland datasets, and adopt sample optimization techniques and more advanced classification algorithms to improve the detection of subpixel terrace distributions.

#### 360 5 Data availability


The Landsat imagery and Copernicus DEM data were acquired from the Google Earth Engine. The GlobeLand30 data can be downloaded from the National Geomatics Center of China. The 1 km spatial resolution SWCTMD (calculated from the 30 m resolution SWCTMD) can be accessed at https://doi.org/10.11888/Terre.tpdc.302400 (Duan, 2025). The 30 m resolution SWCTMD will be available after publication.

#### 365 6 Conclusions



This study developed the first SWC terrace measures dataset for China with a fine classification system at a spatial resolution of 30 m. The dataset includes data for each decade from 2000 to 2020. The dataset was generated by combining the full archive of Landsat imagery, DEM, and nationally scaled samples obtained by manual visualization, using a two-stage random forest classification on the GEE platform. The average OA and average F1 scores for identifying terraces and non-terraces were 91.7% and 88.9%, respectively. For different terrace types, the average OA and F1 scores were 89.4% and 78.9%, respectively.

Compared to existing terrace datasets, the newly developed dataset provides more comprehensive coverage, especially in identifying zig terraces in southwest China. The analysis revealed that terraces were primarily distributed in the Loess Plateau, Southwest China, and Southeast China. From 2000 to 2020, the total terrace areas expanded by 41,594.1 km², with level terraces increasing by 5,701.4 km², slope terraces by 29,876.3 km², slope-separated terraces by 129.9 km², and zig terraces by 5,886.5 km². Terrace expansion was mainly concentrated in the Loess Plateau and the Southwest and Southeast regions of China, while the decreases in terraced area primarily occurred around urban areas.

Terraces in China are estimated to have reduced soil erosion on cropland by approximately 1,390 million tons. Further analysis highlighted the benefits of SWC in the Yunnan-Guizhou Plateau and Loess Plateau areas. The terrace dataset, with its detailed classification system is expected to provide a cornerstone for national and regional soil erosion assessment and prediction, SWC planning, and evaluations of various ecosystem services related to terraces.

## **Author contributions**

XD conceived and designed the study. EZ conducted the construction of the dataset and wrote the manuscript. EZ, HW, BD collected the data. CL provided the technical support. SW, HL, XY and YL provided assistance with the data analysis. YC and XD edited and revised the manuscript.

#### **Competing interests**

The contact author has declared that none of the authors has any competing interests.

#### **Disclaimer**


Publisher's note: Copernicus Publications remains neutral with regard to jurisdictional claims made in the text, published maps, institutional affiliations, or any other geographical representation in this paper. While Copernicus Publications makes every effort to include appropriate place names, the final responsibility lies with the authors. Regarding the maps used in this paper, please note that Figs. 2, 3, 4, 5, 6, and 8 contain disputed territories.

#### Acknowledgments

We express our great gratitude to the free access to the Landsat data provided by the USGS, Copernicus DEM provided under COPERNICUS by the European Union and European Space Agency (ESA), GlobeLand30 provided by the National Geomatics Center of China, and the cloud computing power provided by GEE.

## Financial support

This research was supported by the National Natural Science Foundation Project of China (grant no. U24A20581 and 42271128), the National Key Research and Development Program of China (grant no. 2023YFD1901201), Distinguished Young Found Project of Yunnan Province (grant no. 202201AV070001), the Training Program of the Innovation Guidance and Scientific and Technological Enterprise of Yunnan Province (grant no. 202304BT090019), the Yuanjiang Dry-hot Valley Water and Soil Conservation Observation and Research Station of Yunnan Province.

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
