# Peer review of "A 30 m resolution dataset of soil and water conservation terraces across China for 2000, 2010, and 2020"

_Earth System Science Data, 2025_

## Referee Comment (RC2)

**Comment on essd-2025-215**

This study developed the first nationwide 30 m resolution Soil and Water Conservation Terrace Measures Dataset (SWCTMD) of China from 2000 to 2020. Terrace is the most important type of soil conservation engineering measures for most areas in China, given the critical role of terraces in mitigating soil erosion, enhancing water retention, this work fills a critical data gap that has long constrained national-scale erosion modeling and conservation planning for decades. A two-stage classification framework was proposed, first distinguishing terraces from non-terraces and then classifying four terrace types: level terrace, slope terrace, zig terrace, and slope-separated terrace. The framework integrates time-series Landsat imagery, DEM data, and GlobeLand30 cropland data, employing the Random Forest classifier on the Google Earth Engine platform. Key findings include:

(1)Total terrace area increased from 400,896 km² in 2000 to 496,934 km² in 2020;

(2) Terraces were estimated to reduce about 818 million tons of soil erosion on croplands in 2020;

(3) The dataset outperforms previous products in spatial completeness and terrace type granularity at national scale;

(4) It supports applications in soil erosion modeling, ecosystem service assessment, and land management planning.

**While the study is of high quality and makes a significant contribution, I would like to offer the following suggestions for improvement.**

**Suggestions and Minor Revisions Required :**

(1)Some sentences in the manuscript require more precise wording and would benefit from a more rigorous formulation. For instance, Lines 74-75 currently states:

*"Using this mapping framework, the first Soil and Water Conservation Terrace Measures Dataset of China (SWCTMD) was produced using time-series Landsat satellite imagery and digital elevation model data, covering the period from 2000 to 2020."*

In fact, several nationwide terrace mapping efforts have been conducted prior to this study, such as the work by Cao et al. (2020), which generated a 30-meter resolution terrace map for China. While those studies were not based on long-term time series data, they do represent national-scale mapping efforts. Therefore, the sentence should be revised to clarify that the SWCTMD is the first long-term (two-decade) national terrace dataset, rather than the first national terrace dataset overall. Suggested revision: *"Using this mapping framework, we developed the first long-term (2000 – 2020) national Soil and Water Conservation Terrace Measures Dataset (SWCTMD) of China."*

Lines 370, *Reference: Cao B, Yu L, Naipal V, et al. A 30-meter terrace mapping in China using Landsat 8 imagery and digital elevation model based on the Google Earth Engine. Earth System Science Data Discussions, 2020: 1 – 35.*

Similarly, *Lines 108-109 "Compared to other DEM data, SRTM DEM is the most quality-controlled, broadest coverage, and highest accuracy DEM among open-source data"*

While SRTM has historically been one of the most widely used and quality-controlled global DEM datasets, it is worth noting that in recent years, several newer open-access DEM products, such as ALOS AW3D30 and Copernicus DEM, have demonstrated higher spatial resolution and improved accuracy in many regions, including mountainous and vegetated areas.

(2)According to the manuscript, all four types of terraces used in the study are assigned dimensionless

conservation factor (P or E-values) below 0.343. This implies that, under the RUSLE/CSLE framework, terrace implementation on a given slope would reduce potential soil loss by at least 65.7%. Furthermore, since terraces are predominantly distributed on sloping croplands, areas generally subject to higher erosion rates, the expected relative reduction in soil loss should arguably be even more significant when terraces are applied. However, In Section 3.4, the authors state that, "In comparison to the scenario without terrace measures, the amount of soil erosion in the regions of Yunnan, Sichuan, Chongqing, Guizhou, Gansu, Shanxi, and Shaanxi regions decreased by 47.47%, 46.02%, 45.57%, 45.25%, 35.48%, 29.75%, and 27.80%, respectively (Fig. 8b)". Therefore, I would appreciate it if the authors could clarify the reason behind.

(3)While the overall accuracy (OA) and Kappa coefficient reported for the classification results appear high, this does not necessarily imply satisfactory classification performance. Notably, the Producer's Accuracy (PA) for *zig terraces* and *level terraces* was particularly low (as low as 15–25%), and their corresponding F1-scores were generally below 40%, indicating considerable misclassification and omission errors. Such results suggest that these terrace types may have been substantially underestimated in the dataset. It is important to note that high OA/Kappa values may be misleading in imbalanced classification tasks, especially when majority classes dominate the confusion matrix. The low performance metrics for certain terrace types likely reflect a combination of factors, including:

> *Class imbalance between dominant and minority categories during model training;*
> *Limited or unrepresentative training samples for fragmented or narrow terraces;*
> *Intrinsic heterogeneity in real-world spatial distribution, especially in mountainous regions;*
> *Fragmented and narrow terraces are especially prone to omission;*
> *Spectral confusion and mixed pixels in medium-resolution imagery.*

These aspects should be discussed in the manuscript to provide a more nuanced interpretation of the classification results and to guide future refinement. The authors may also refer to relevant studies for in-depth discussions on these issues.

(4) Since the manuscript involves the quantitative assessment of soil and water conservation benefits of terraces, particularly through the estimation of soil erosion reduction, it is important to critically examine the assumption of using a uniform conservation factor (P-value or E-value) for the same type of terrace measures across different regions of China. While assigning a single value to each terrace type simplifies the model and facilitates national-scale analysis, it may overlook important spatial heterogeneity in terrace structure, maintenance conditions, climatic regimes, and land management practices. For example:

*The same "slope terrace" may perform differently in terms of erosion control in Yunnan's highlands compared to the Loess Plateau, due to differences in rainfall erosivity, soil properties, and vegetation cover; Engineering design standards and actual field implementation of terraces may vary significantly between provinces, leading to divergence in functional effectiveness; The topographic context (e.g., slope gradient, curvature) strongly influences the conservation outcome, even for structurally similar terraces.*

Therefore, I recommend that the authors acknowledge and briefly discuss the limitations of using fixed terrace factor values in erosion modeling across diverse environmental settings.

---

## Author Response (AR2)

MS No.: essd-2025-215

MS Type: Data description paper

Title: A 30 m soil and water conservation terrace measures dataset of China from 2000 to 2020

Dear Editors and Reviewers,

On behalf of my co-authors, we would like to express our sincere gratitude to the editor and the reviewers for your constructive feedback and thorough review of our manuscript. We have carefully considered all suggestions and have made the corresponding revisions to the manuscript. Consequently, our manuscript

has been considerably improved.

Below, we provide detailed responses to the reviewer's comments, including clarifications where necessary. We hope these revisions address the concerns and uncertainties raised by the reviewer. The revised version with tracked changes is provided as the "Revision changes marked" file and the clean version with all changes accepted is provided as the "Manuscript" file. The line and figure numbers are based on the "Manuscript" file. All the changes were highlighted (red color) in the "Revision changes marked" file. A point-by-point response to the comments of the reviewers is also listed below.

Sincerely,

Xingwu Duan, on behalf of all co-authors

xwduan@ynu.edu.cn

Yunnan Key Laboratory of Soil Erosion Prevention and Green Development, Institute of International

Rivers and Eco-security, Yunnan University

**Comments from Reviewer #1:**

Terracing is one of the most important soil and water conservation (SWC) measures in China, playing a critical role in mitigating soil erosion. This study proposes a two-stage mapping framework to classify different types of terrace measures and develops a new dataset—the Soil and Water Conservation Terrace Measures Dataset (SWCTMD)—based on time-series Landsat imagery and digital elevation model (DEM) data from 2000 to 2020. The framework incorporates a refined classification system that provides detailed information on both terrace distribution and associated SWC measure factors, offering significant value for understanding and managing soil erosion dynamics.

Response: We greatly appreciate your positive comments concerning our manuscript. Those comments are valuable and helpful for improving our manuscript. We followed all comments and made revision and responses carefully. A point-by-point reply to the comments are listed below.

1. It is recommended to revise the title to: A 30 m resolution dataset of soil and water conservation terraces across China (2000–2020).

Response: Thank you very much for your valuable suggestion regarding the title. Based on all reviewers' comments on the title, we have revised the title to "A 30 m resolution dataset of soil and water conservation terraces across China for 2000, 2010, and 2020" (Lines 1-2).

2. What specific subcategories of cropland are included in the study?

Response: The cropland data from GlobeLand30 used in this study does not have a secondary classification. The cropland is defined as land used for cultivating crops, including irrigated farmlands, paddy fields, green houses cultivated land, artificial tame pastures, economic cultivated land (e.g., grape, coffee, and palm), and abandoned arable lands.

3. What information is contained in the 30 m grid dataset? Does it include whether the area is terraced or not, the type of terrace, and the associated conservation measure factor?

Response: The 30-meter resolution Soil and Water Conservation Terrace Measures Dataset (SWCTMD) includes maps of different terrace types, along with corresponding values of soil and water conservation measure factors for each type.

4. It is recommended to retain numerical values to one decimal place for better clarity and consistency.

Response: Thank you for the valuable suggestion. We revised the entire manuscript, retaining numerical values to one decimal place.

5. In Table 2, the four types of terrace measures are shown with remote sensing images. To enhance

clarity and visual recognition, it is recommended to replace these with high-resolution photographs.

Response: Thank you for your insightful comment. I have replaced the remote sensing images with high-resolution field photographs.

Table 2. Image characteristics of different terrace types.

| Terrace types                  | Image characteristics                                                                                                                                                                                                                                                                                                        | Remote sensing image |
|--------------------------------|------------------------------------------------------------------------------------------------------------------------------------------------------------------------------------------------------------------------------------------------------------------------------------------------------------------------------|----------------------|
| Level terrace                  | Steep slope land transformed into a series of successively receding flat surfaces, with bunds constructed from soil or stones, ranging in width from 5 to 40 m, looking like the steps of a staircase in remote sensing images. In contrast to slope terraces, level terraces are predominantly found in low and flat areas. |                      |
| Slope terrace                  | Similar to level terraces, but with wider and more uneven surfaces, these terraces exhibit irregular shapes in remote sensing images. They are primarily used for dryland agriculture and are largely distributed the areas with slopes greater than 5°.                                                                     |                      |
| Zig terrace                    | Steep slope land has been transformed into step-like terraces that are narrower than level terraces. The surfaces of these terraces exhibit regular strip shapes in remote sensing images. These terraces are primarily found in sloping regions and are used for planting permanent crops such as tea.                      |                      |
| Slope-
separated
terrace | Each flat surface constructed on steep slope land retains a segment of the original slope above, forming a composite structure that features a slope between flat surfaces. These terraces are primarily used for rubber plantations.                                                                                        |                      |

6. In the accuracy assessment section, the evaluation metrics should be further explained, such as the possible value ranges of each indicator and whether higher or lower values indicate better accuracy.

Response: Thank you for the constructive suggestion. We have added the relevant explanations in Section 2.8. The revised sentence in is provided in Lines 179-184:

"... This method offers quantitative assessment metrics, including the kappa coefficient (KA), overall accuracy (OA), the producer's accuracy (PA), and the user's accuracy (UA), which collectively assess the performance of the products. OA and KA measure the total map accuracy. PA and UA measure the omission and commission errors for each class. In addition, we calculated the F1 score, which reflects the balance between UA and PA. The KA, OA, PA, UA, and F1 metrics range from 0 to 1, where 1 indicates optimal performance and 0 represents the poorest performance..."

7. The methodology section lacks details about the estimation of soil erosion and the other influencing factors used in the analysis related to terrace responses in China. Please specify the estimation methods

and data sources for these additional factors.

Response: Thank you very much for pointing out this issue. In the supplementary materials, we have added the evaluation method for soil erosion in China (Supplementary Note S1), the calculation approach for erosion area (Supplementary Note S2), and the analytical method for assessing the impact of terracing on soil erosion (Supplementary Note S3). Additionally, we have clearly specified the data sources and the relevant preprocessing steps.

8. In Figure 3, should the legend indicate "cropland with terraces" and "cropland without terraces," rather than "cropland" and "terraces"? Additionally, does the map on the left represent the distribution of cropland? Please clarify it.

Response: Thanks for pointing out this omission. Here, the legends for the two subgraphs are placed together. The map on the left side of figure 4 represents the distribution of cropland in China in 2020 (Figure 4a). The map on the right side of Figure 4 shows the spatial distribution of the three terraces datasets (Figure 4b). We have created legends for Figure 4a and Figure 4b respectively. Figure 4 has been updated and the legend meanings have been clarified:

Figure 5. The spatial patterns of different terrace types at the pixel and provincial. (a) The spatial distribution of different terraces in China in 2020. (b) The different terrace areas in provinces in 2020.

9. To highlight the novelty of this dataset, it is recommended to include a comparison table in addition to Figure 3. This table should provide quantitative details of the differences between this new dataset and existing ones, such as the extent of area change in specific regions and the types of terraces contributing to those changes.

Response: Thank you for your helpful comment. We have added two tables to the supplementary materials, namely Table S8 and Table S9. Table S8 presents the differences in provincial terrace areas

between SWCTMD and CTM2018, along with the types of terraces contributing to those changes. In the Section 3.1, we revised the sentences (Lines 203-212). Furthermore, to analyze the differences among the three datasets, In the Section 4.1, we compared terrace areas at the provincial scale and analyzed the causes of these discrepancies. The revised text now stands as follows (Lines 278-288):

"Figure 4 illustrates the spatial consistency between the SWCTMD and two existing datasets: the 2018 China Terrace Map (CTM2018) (Cao et al., 2021) and the 2017 China Terrace Map (CTM 2017) (Li et al., 2024). SWCTMD exhibited the highest accuracy. Compared to SWCTMD and CTM2018, CTM2017 exhibited relatively lower accuracy for both typical terrace and non-terraces areas (regions A, B, C, D F and G in Fig. 4b). For typical terraces, SWCTMD and CTM2018 show similar identification performance (regions A, B, C and F in Fig. 4b). However, for atypical terraces, such as zig terraces located in Yunnan Province, SWCTM successfully identified these as terraces, whereas CTM2018 failed to identify them as terraces (regions E in Fig. 4b). Conversely, for non-terrace areas situated in the Middle-Lower Yangtze River, SWCTMD accurately classified these as non-terraces, while CTM2018 erroneously classified them as terrace areas (regions G in Fig. 4b). At the provincial scale, the majority of provinces exhibit larger terrace areas in SWCTMD compared to both CTM2018 and CTM2017 (Tables S8 and S9)."

"We compared the 2020 terrace area estimated by SWCTMD with those from CTM2018 and CTM2017. SWCTMD exhibited the largest terrace area compared to CTM2018 and CTM2017. The areal discrepancies can be attributed to the following reasons. First, CTM2017 and CTM2018 predominantly focused on the most typical level terraces, whereas our research encompasses a broader range of terrace types, including non-typical terraces such as slope terraces, zig terraces, and slope-separated terrace. Second, each dataset employed distinct cropland for terrace classification. SWCTMD utilized the union of cropland with slopes exceeding 2° from the 2000, 2010, and 2020 GlobeLand30 cropland data, whereas CTM2018 employed only the 2010 GlobeLand30 cropland data, and CTM2017 adopted FROM-GLC cropland data. Third, CTM2018 excluded isolated patches smaller than 9,000 m² from its classification scheme. However, since SWCTMD constrains its classification to cropland with slopes exceeding 2°, the identified terrace areas in Anhui, Fujian, Jiangxi, and Zhejiang provinces were smaller than those from CTM2018. In these provinces, CTM2018 included terraces with slopes below 2°, which is classified as non-terraces according to the technical regulations of the third nationwide land survey. Overall, our dataset provides more comprehensive coverage for terraces and exhibits higher accuracy and robustness."

Table S8. Comparison of terrace Areas between SWCTMD in 2020 and CTM2018 Datasets. "Gain' denotes areas identified as terraces only in the SWCTMD dataset, whereas "Loss" refers to areas identified as terraces only in the CTM2018 dataset.

|          |                  | SWCTMD           | Difference                     | Gain (km²        | 2)               |                |                                | Loss (km²) |
|----------|------------------|------------------|--------------------------------|------------------|------------------|----------------|--------------------------------|------------|
| province | CTM2018
(km²) | in 2020
(km²) | (SWCTMD –
CTM2018)
(km²) | Level
terrace | Slope
terrace | Zig
terrace | Slope-
separated
terrace | Terrace    |
| Anhui    | 7255.19          | 5092.39          | -2162.80                       | 2414.14          | 325.61           | 60.99          | /                              | 4963.55    |
| Fujian   | 9097.75          | 6637.96          | -2459.79                       | 1002.26          | 510.44           | 479.38         | /                              | 4451.87    |

| Gansu     | 52248.06 | 57356.15 | 5108.09  | 8007.46  | 1091.93  | 339.48  | /      | 4330.79  |
|-----------|----------|----------|----------|----------|----------|---------|--------|----------|
| Guangdong | 4982.90  | 6597.43  | 1614.52  | 2051.30  | 996.20   | /       | /      | 1432.97  |
| Guangxi   | 13133.09 | 20975.42 | 7842.33  | 8423.77  | 4339.73  | /       | /      | 4921.16  |
| Guizhou   | 46315.57 | 59035.48 | 12719.91 | 9192.84  | 4410.92  | 1226.73 | 6.24   | 2116.83  |
| Hebei     | 13751.55 | 15209.87 | 1458.32  | 2853.21  | 2320.73  | /       | 0.00   | 3715.62  |
| Henan     | 14027.56 | 23705.55 | 9678.00  | 7959.63  | 3491.49  | /       | 17.69  | 1790.81  |
| Hubei     | 23267.58 | 26391.32 | 3123.74  | 7600.61  | 3458.52  | /       | 27.13  | 7962.52  |
| Hunan     | 30868.80 | 40813.88 | 9945.08  | 10964.49 | 5169.61  | /       | 9.15   | 6198.18  |
| Jiangxi   | 15564.45 | 7512.25  | -8052.20 | 2181.49  | 391.19   | 296.52  | /      | 10921.41 |
| Ningxia   | 8035.61  | 10194.15 | 2158.53  | 2247.29  | 79.69    | 141.31  | /      | 309.76   |
| Shandong  | 18302.50 | 23123.89 | 4821.39  | 5771.31  | 1584.79  | 46.37   | /      | 2581.08  |
| Shanxi    | 37062.87 | 41679.43 | 4616.57  | 6286.92  | 2595.84  | 0.00    | /      | 4266.19  |
| Shaanxi   | 30977.01 | 39467.78 | 8490.77  | 7044.01  | 4313.44  | 75.73   | /      | 2942.41  |
| Sichuan   | 83417.53 | 99285.71 | 15868.18 | 13574.22 | 7424.91  | 475.86  | 4.31   | 5611.11  |
| Yunnan    | 75073.95 | 97955.88 | 22881.93 | 6853.27  | 12469.88 | 7584.56 | 324.43 | 4350.22  |
| Zhejiang  | 6334.11  | 3807.41  | -2526.70 | 743.45   | 305.45   | 94.45   | /      | 3670.05  |
| Chongqing | 29746.65 | 38039.26 | 8292.62  | 5864.75  | 3569.64  | 479.07  | 2.59   | 1623.43  |
| Qinghai   | 6038.90  | 6316.37  | 277.47   | 792.53   | 88.16    | 47.56   | 0.00   | 650.78   |
|           |          |          |          |          |          |         |        |          |

Table S9. Comparison of terrace Areas between SWCTMD in 2020 and CTM2017 Datasets. "Gain" denotes areas identified as terraces only in the SWCTMD dataset, whereas "Loss" refers to areas identified as terraces only in the CTM2017 dataset.

|           |          | SWCTMD           | Difference (SWCTMD | Gain (km²)    |                  |                |                                | Loss
(km²) |
|-----------|----------|------------------|--------------------|---------------|------------------|----------------|--------------------------------|---------------|
| province  |          | in 2020
(km²) | - CTM) (km²)       | Level terrace | Slope
terrace | Zig
terrace | Slope-
separated
terrace | Terrace       |
| Anhui     | 2320.54  | 5093.47          | 2772.93            | 3935.96       | 448.74           | 74.63          | /                              | 1686.40       |
| Fujian    | 6698.71  | 6636.65          | -62.06             | 2749.31       | 867.76           | 686.97         | /                              | 4366.09       |
| Gansu     | 32161.56 | 57353.72         | 25192.16           | 26441.27      | 4092.22          | 2106.95        | /                              | 7448.28       |
| Guangdong | 4208.52  | 6594.76          | 2386.24            | 3697.91       | 1261.08          | /              | /                              | 2572.75       |

|           |                            | SWCTMD   | Difference (SWCTMD | Gain (km²        | )              |                                |         | Loss
(km²) |
|-----------|----------------------------|----------|--------------------|------------------|----------------|--------------------------------|---------|---------------|
| province  | province CTM2017 in 2020 \ | - CTM)   | Level
terrace   | Slope
terrace | Zig
terrace | Slope-
separated
terrace | Terrace |               |
| Guangxi   | 7308.95                    | 20976.30 | 13667.35           | 12190.39         | 5446.80        | 0.00                           | /       | 3969.85       |
| Guizhou   | 25456.48                   | 59035.79 | 33579.31           | 27157.62         | 10924.35       | 2351.29                        | 11.84   | 6865.79       |
| Hebei     | 13537.45                   | 15210.61 | 1673.16            | 4921.16          | 3876.86        | /                              | /       | 7124.85       |
| Henan     | 13963.48                   | 23708.82 | 9745.34            | 9837.58          | 4299.97        | /                              | 49.04   | 4441.25       |
| Hubei     | 14110.44                   | 26391.38 | 12280.93           | 13296.29         | 5157.65        | /                              | 41.99   | 6215.00       |
| Hunan     | 27467.65                   | 40811.43 | 13343.79           | 18362.07         | 6781.28        | /                              | 23.45   | 11823.01      |
| Jiangxi   | 9068.41                    | 7510.90  | -1557.51           | 4355.89          | 581.50         | 409.47                         | /       | 6904.37       |
| Ningxia   | 3994.61                    | 10195.18 | 6200.57            | 6455.23          | 158.86         | 755.13                         | /       | 1168.64       |
| Shandong  | 18097.66                   | 23123.08 | 5025.42            | 6757.42          | 2390.43        | 420.78                         | /       | 4543.20       |
| Shanxi    | 29853.48                   | 41677.98 | 11824.50           | 15744.13         | 5903.42        | /                              | /       | 9823.05       |
| Shaanxi   | 28370.55                   | 39467.41 | 11096.85           | 14514.39         | 7367.91        | 391.46                         | /       | 11176.91      |
| Sichuan   | 59470.14                   | 99284.94 | 39814.80           | 36890.96         | 14167.60       | 781.83                         | 7.66    | 12033.25      |
| Yunnan    | 38251.80                   | 97945.82 | 59694.02           | 19271.94         | 38146.24       | 13171.54                       | 347.91  | 11243.60      |
| Zhejiang  | 2780.66                    | 3807.41  | 1026.75            | 2102.36          | 624.01         | 202.83                         | /       | 1902.45       |
| Chongqing | 20357.14                   | 38039.24 | 17682.10           | 13703.93         | 6753.94        | 688.10                         | 4.21    | 3468.08       |
| Qinghai   | 5519.58                    | 6315.62  | 796.03             | 1999.26          | 465.99         | 281.98                         | /       | 1951.19       |

10. In Figure 6, it is suggested to label the numerical values within each grid cell to improve readability and interpretability.

Response: We thank the reviewer for making this helpful suggestion. We have added numerical labels within each grid cell to enhance readability and interpretability. Figure 7 has been updated:

Figure 7. The changes of terrace areas at provincial and types from 2000 to 2020. (a) The changes of terrace area in different provinces. (b) The proportion of terraces to cropland in different provinces. (c-f) The areas of level terrace, slope terrace, zig terrace, and slope-separated terrace. (g) The total terrace areas of China.

11. Many quantitative descriptions in the manuscript only provide absolute values; it is recommended to also include relative percentages to help readers better interpret the significance of the results.

Response: Thank you for the valuable suggestion. We carefully reviewed the manuscript and, in quantitative descriptions, provided both absolute values and relative percentages to help readers better understand the significance of the results.

Accordingly, the sentences (Lines 251-257) have been revised as follows:

"... From 2000 to 2020, Yunnan, Guangxi, Shanxi, and Shaanxi experienced the most significant increases in terrace areas, increasing by 11,372.4 km² (13.1%), 5,192.4 km² (32.9%), 2,395 km² (6.1%), and 2,295.0 km² (6.2%), respectively (Fig. 7a). In terms of terrace types, the areas of level terraces, slope terraces, zig terraces and slope separated terraces increased by 5,701.4 km² (1.3%), 29,876.3 km² (18.9%), 5,886.5 km² (31.4%), and 129.9 km² (24.9%), respectively, with the slope terrace having the largest increase (Figs. 7c, d, e and f). Overall, China's total terrace area expanded from 612,885.4 km² in 2000 to 654,479.5 km² in 2020, an increase of 6.8% (Fig. 7g).

12. In Figure 7, are fixed values assigned to each type of terrace measure? Please clarify the value assignment approach.

Response: Thank you for this important comment. In soil erosion assessment, the soil and water conservation engineering practice factor (E) is generally assigned by a value to each measure. In this study, the E values for different terrace types were assigned according to the China's First National

Census for Water and published literature (Table S12) (Duan et al., 2020; Liu et al., 2020).

Table S12. Values of E factor.

| Level terrace | Slope terrace | Zig terrace | Slope-separated terrace |
|---------------|---------------|-------------|-------------------------|
| 0.01          | 0.252         | 0.114       | 0.343                   |

**References**

Duan, X., Bai, Z., Rong, L., Li, Y., Ding, J., Tao, Y., Li, J., Li, J., and Wang, W.: Investigation method for regional soil erosion based on the Chinese Soil Loss Equation and high-resolution spatial data: Case study on the mountainous Yunnan Province, China, Catena, 184, 104237, https://doi.org/10.1016/j.catena.2019.104237, 2020.

Liu, B., Xie, Y., Li, Z., Liang, Y., Zhang, W., Fu, S., Yin, S., Wei, X., Zhang, K., Wang, Z., Liu, Y., Zhao, Y., and Guo, Q.: The assessment of soil loss by water erosion in China, Int. Soil Water Conserv. Res., 8, 430–439, https://doi.org/10.1016/j.iswcr.2020.07.002, 2020.

13. The sentence: "According to our estimation, the soil erosion of the Loess Plateau accounts for only 10.95% of the total cropland erosion in China, indicating that the SWC measures previously implemented have achieved good governance..." is better rephrased with the emphasis placed on the comparative effects of having terraces vs. not having terraces, or the differences among this dataset and previpus datasets, in order to reflect the value of this dataset in soil erosion estimation. The relatively low share of cropland erosion in the Loess Plateau does not necessarily indicate the effectiveness of conservation measures alone—it may also be influenced by factors such as total cropland area, topography, vegetation cover, climate and so on.

Response: Thank you for your insightful comment. Sorry for the unclear expression. We have revised this sentence (Lines 323-325). We shifted the focus of the description to the amount of soil erosion reduced by terraces, emphasizing the impact of terraces on soil erosion. The updated version reads:

"According to our estimate, soil erosion of the Loess Plateau accounted for only 12.6% of the total cropland erosion. Terraces in this region contributed to 17.4% of the total reduction of cropland soil erosion, demonstrating the benefits of terraces to SWC. ..."

**Comments from Reviewer #2:**

This study developed the first nationwide 30 m resolution Soil and Water Conservation Terrace Measures Dataset (SWCTMD) of China from 2000 to 2020. Terrace is the most important type of soil conservation engineering measures for most areas in China, given the critical role of terraces in mitigating soil erosion, enhancing water retention, this work fills a critical data gap that has long constrained national-scale erosion modeling and conservation planning for decades. A two-stage classification framework was proposed, first distinguishing terraces from non-terraces and then classifying four terrace types: level terrace, slope terrace, zig terrace, and slope-separated terrace. The framework integrates time-series Landsat imagery, DEM data, and GlobeLand30 cropland data, employing the Random Forest classifier on the Google Earth Engine platform. Key findings include:

- (1) Total terrace area increased from 400,896 km2 in 2000 to 496,934 km2 in 2020;
- (2) Terraces were estimated to reduce about 818 million tons of soil erosion on croplands in 2020;
- (3) The dataset outperforms previous products in spatial completeness and terrace type granularity at national scale;
- (4) It supports applications in soil erosion modeling, ecosystem service assessment, and land management planning.

While the study is of high quality and makes a significant contribution, I would like to offer the following suggestions for improvement.

Suggestions and Minor Revisions Required:

Response: We appreciate you very much for your positive comments concerning our manuscript. Those comments are valuable and helpful for improving our manuscript. We followed all comments and made revision and responses carefully. A point-by-point reply to the comments are listed below.

1. Some sentences in the manuscript require more precise wording and would benefit from a more rigorous formulation. For instance, Lines 74-75 currently states:

"Using this mapping framework, the first Soil and Water Conservation Terrace Measures Dataset of China (SWCTMD) was produced using time-series Landsat satellite imagery and digital elevation model data, covering the period from 2000 to 2020."

In fact, several nationwide terrace mapping efforts have been conducted prior to this study, such as the work by Cao et al. (2020), which generated a 30-meter resolution terrace map for China. While those studies were not based on long-term time series data, they do represent national-scale mapping efforts. Therefore, the sentence should be revised to clarify that the SWCTMD is the first long-term (two-decade) national terrace dataset, rather than the first national terrace dataset overall. Suggested revision: "Using this mapping framework, we developed the first long-term (2000–2020) national Soil and Water Conservation Terrace Measures Dataset (SWCTMD) of China."

Lines 370, Reference: Cao B, Yu L, Naipal V, et al. A 30-meter terrace mapping in China using Landsat

8 imagery and digital elevation model based on the Google Earth Engine. Earth System Science Data Discussions, 2020: 1–35.

Similarly, Lines 108-109 "Compared to other DEM data, SRTM DEM is the most quality-controlled, broadest coverage, and highest accuracy DEM among open-source data"

While SRTM has historically been one of the most widely used and quality-controlled global DEM datasets, it is worth noting that in recent years, several newer open-access DEM products, such as ALOS AW3D30 and Copernicus DEM, have demonstrated higher spatial resolution and improved accuracy in many regions, including mountainous and vegetated areas.

Response: Thank you for your insightful comment. We apologize for the unclear expression. The original sentence was: "Using this mapping framework, the first Soil and Water Conservation Terrace Measures Dataset of China (SWCTMD) was produced using time-series Landsat satellite imagery and digital elevation model data, covering the period from 2000 to 2020." We revised it to the suggested version: "Using this mapping framework, we developed the first long-term (2000 to 2020) national Soil and Water Conservation Terrace Measures Dataset (SWCTMD) of China." (Lines 75-77).

We recognized that several newer open-source DEM data products in recent years offer higher accuracy than SRTM DEM, particularly in mountainous and vegetated areas. Therefore, we replaced the SRTM DEM data with Copernicus DEM data and regenerated the SWCTMD dataset.

Accordingly, the sentences (Lines 105-110) have been revised as follows:

"Topographical features are essential characteristics that differentiate regular cropland and terrace, playing a crucial role in the identification of terraces. We used the Copernicus DEM data to calculate these topographical features. The Copernicus DEM is a Digital Surface Model with 30 m resolution, derived from radar satellite data acquired from 2010 to 2015 during the TanDEM-X mission. Compared to other DEM data (SRTM, ASTER GDEM, ALOS World 3D, and NASADEM), Copernicus DEM has the highest accuracy among open-source data (Guth and Geoffroy, 2021), exhibiting the greatest detail of terrain (Li et al., 2022a). The GEE platform provides access to the Copernicus DEM at 30 m resolution."

2. According to the manuscript, all four types of terraces used in the study are assigned dimensionless conservation factor (P or E-values) below 0.343. This implies that, under the RUSLE/CSLE framework, terrace implementation on a given slope would reduce potential soil loss by at least 65.7%. Furthermore, since terraces are predominantly distributed on sloping croplands, areas generally subject to higher erosion rates, the expected relative reduction in soil loss should arguably be even more significant when terraces are applied. However, In Section 3.4, the authors state that, "In comparison to the scenario without terrace measures, the amount of soil erosion in the regions of Yunnan, Sichuan, Chongqing, Guizhou, Gansu, Shanxi, and Shaanxi regions decreased by 47.47%, 46.02%, 45.57%, 45.25%, 35.48%, 29.75%, and 27.80%, respectively (Fig. 8b)". Therefore, I would appreciate it if the authors could clarify the reason behind.

Response: We assigned a dimensionless conservation factor (P or E value) below 0.343 for each type of terrace. Under the RUSLE/CSLE framework, terrace implementation on a given slope would reduce potential soil loss by at least 65.7%. However, this reduction is achievable only when the terrace area is close to or approximately equal to the cropland area. For instance, when the terrace area in a province is

significantly smaller than its cropland area, the total amount of soil erosion reduced by terraces is limited. Compared to scenarios without terrace measures, the proportion of soil erosion reduction would be far below 65.7%. Furthermore, the soil and water conservation effectiveness of terraces also depends on their spatial distribution. If a province's terraces are mainly distributed in areas with moderate—and strong soil erosion intensity, with few in areas of very strong and severe soil erosion intensity. Since the region with very strong and severe soil erosion intensity may have contributed to the majority of the province's total soil erosion, the proportion of soil erosion reduction would be also far below 65.7%.

3. While the overall accuracy (OA) and Kappa coefficient reported for the classification results appear high, this does not necessarily imply satisfactory classification performance. Notably, the Producer's Accuracy (PA) for zig terraces and level terraces was particularly low (as low as 15–25%), and their corresponding F1-scores were generally below 40%, indicating considerable misclassification and omission errors. Such results suggest that these terrace types may have been substantially underestimated in the dataset. It is important to note that high OA/Kappa values may be misleading in imbalanced classification tasks, especially when majority classes dominate the confusion matrix. The low performance metrics for certain terrace types likely reflect a combination of factors, including:

Class imbalance between dominant and minority categories during model training;

Limited or unrepresentative training samples for fragmented or narrow terraces;

Intrinsic heterogeneity in real-world spatial distribution, especially in mountainous regions;

Fragmented and narrow terraces are especially prone to omission;

Spectral confusion and mixed pixels in medium-resolution imagery.

These aspects should be discussed in the manuscript to provide a more nuanced interpretation of the classification results and to guide future refinement. The authors may also refer to relevant studies for in-depth discussions on these issues.

Response: Thank you for your insightful comment. In the revised manuscript Section 4.3, we discussed the impact of sample imbalance, uneven distribution of terraces, spectral confusion, and mixed pixels on classification accuracy, and proposed recommendations for improving the classification of terraces in the future. It now stands as follows (Lines 464-489):

"The spatial heterogeneity of land types frequently leads to class imbalance in remote sensing classification, consequently diminishing classification accuracy for minority classes that occupy a smaller area (Xiao et al., 2024). The models tend to favor majority classes during training, reducing their ability to accurately identify minority classes (Chen et al., 2025). When the ratio of samples across different classes remains balanced, classification performance typically falls short of optimal accuracy thresholds (Deng et al., 2025). A common strategy to alleviate this negative effect is to divide the study area into multiple sub-regions for localized classification, thereby reducing the impact of sample imbalance on model accuracy (Zhang et al., 2020). In this study, we employed a partitioned two-stage RF approach to reduce the effects of sample imbalance on classification accuracy. The results demonstrated that classification for terrace and different terrace types achieved satisfactory accuracy in both the entire study area and individual subregions. However, the accuracy metrics of the majority class

were still higher than those of the minority class. In future studies, sample optimization techniques and more advanced classification methods could be combined to further improve the accuracy of minority class classification."

"The complex and diversity diverse landform types have resulted in differences in the spectral information and topographic features of terraces in different regions. In Southwest and Northwest China, terraces exhibit concentrated distributions with clearly defined characteristics, making them easily identifiable. However, South China, Central China, and East China have relatively low topographic relief. Some terraces have spectral and topographic features similar to those of sloping farmland and flatland. This similarity, combined with the presence of mixed pixels in medium-resolution imagery (Wang et al., 2021a), makes it challenging to detect the terrace patches. Although the classification used 30-meter Landsat imagery in this study was generally robust, some fragmented and narrow terraces were omitted. Future research could employ high-resolution remote sensing images to effectively identify fragmented and narrow terraces. Previous small-scale studies have demonstrated that the use of high-resolution remote sensing imagery, combined with object-based classification methods and deep learning approaches, can significantly enhance classification accuracy and reduce the impact of spectral confusion and mixed pixels on terrace identification (Diaz-Varela et al., 2014; Wang et al., 2023; Kan et al., 2025). To improve classification accuracy and efficiency, cropland data were used as the basis for terrace identification. Inevitably, the accuracy of cropland data impacts the terrace mapping process, as errors in cropland data propagate into the terrace maps. In summary, future studies could utilize highresolution remote sensing imagery and more accurate cropland datasets, and adopt sample optimization techniques and more advanced classification algorithms to improve the detection of subpixel terrace distributions."

4. Since the manuscript involves the quantitative assessment of soil and water conservation benefits of terraces, particularly through the estimation of soil erosion reduction, it is important to critically examine the assumption of using a uniform conservation factor (P-value or E-value) for the same type of terrace measures across different regions of China. While assigning a single value to each terrace type simplifies the model and facilitates national-scale analysis, it may overlook important spatial heterogeneity in terrace structure, maintenance conditions, climatic regimes, and land management practices. For example:

The same "slope terrace" may perform differently in terms of erosion control in Yunnan's highlands compared to the Loess Plateau, due to differences in rainfall erosivity, soil properties, and vegetationcover; Engineering design standards and actual field implementation of terraces may vary significantly between provinces, leading to divergence in functional efectiveness; The topographic context (e.g., slope gradient, curvature) strongly influences the conservation outcome, even for structurally similar erraces.

Therefore, I recommend that the authors acknowledge and briefly discuss the limitations of using fixed terrace factor values in erosion modeling across diverse environmental settings.

Response: Thank you for your insightful comment. We recognize the limitations of using fixed terrace factor values in erosion modeling across diverse environmental settings. We discuss this in Section 4.3.

It now stands as follows (Lines329-332):

"In this study, each terrace type was assigned a fixed E value to facilitate the estimation of large-scale soil erosion. However, this approach overlooks spatial heterogeneity in terrace structure, maintenance status, field management, climate, and topography. Future research should incorporate regional characteristics and adjust the E-value accordingly."

**Comments from Reviewer #3:**

This paper aims to provide a dataset of multi-type terraces in China from 2000 to 2020. However, this study primarily focuses on the practical applications of the dataset (many contents related to spatial-temporal analysis and variable importance evaluation), while current methods have obvious shortcomings in terms of data applicability, model robustness, and validation rigor, which affect the credibility of the datasets. This paper may be more appropriate for agricultural journals than for dataset publication. Please see specific comments:

Response: Thank you for these insightful comments and the opportunity to improve our manuscript. We agree that the credibility of a dataset is paramount. In our revision, we have significantly strengthened the methodological rigor, validation, and discussion of the dataset itself to address your concerns directly. We followed all comments and made revisions and responses carefully. A point-by-point reply to the comments are listed below.

Addressing methodological limitations: (1) Applicability. To ensure that our mapping framework is applicable across diverse landscapes in China, we trained the models separately for different regions to map terraces. This strategy guarantees reliable performance in China's highly heterogeneous terrain. (2) Robustness. Our regionally stratified two-stage mapping framework effectively mitigates the impact of sample imbalance and the challenges of classifying different terrace types, thereby improving classification accuracy. We conducted a comprehensive validation using 14,986 high-quality field survey samples derived from China's First National Census for Water. The terrace classification achieved an average overall accuracy (OA) of 91.7% and an average F1-score of 83.4%. For different terrace types, the average overall classification accuracy reached 89.4% with an average F1-score of 78.9%.

In addition, we have restructured the manuscript to highlight that the primary contribution lies in the dataset itself (SWCTMD), rather than in the spatiotemporal analysis. For example, we removed unnecessary sections (Section 3.3 in the original version), added new content in Sections 3.2 and 4.1, and revised Section 4.4 to strengthen the evaluation of data quality. The application analyses (such as soil erosion reduction) are now presented solely to demonstrate the practical utility and reliability of the dataset, which constitutes essential criteria for a data-oriented publication.

We believe that these comprehensive revisions have aligned the manuscript closely with the aims of Earth System Science Data (ESSD). The journal for the publication of articles on original research data (sets), furthering the reuse of high-quality data of benefit to Earth system sciences. The SWCTMD dataset provides a critical and previously unavailable input for Earth system modeling, and its release will enable more accurate assessments of soil erosion, the carbon cycle, and agricultural sustainability.

1. The author primarily utilized 30-meter resolution imagery and DEM. However, many terrace surfaces are smaller than 30 meters. It is uncertain whether these data can accurately represent the objects and extent of multi-type terraces. This implies that there are numerous areas with mixed pixel issues. Additionally, especially for the Zig terrace, which have relatively steep slopes (as stated in Table 2), 30-meter data is insufficient to distinguish such objects. I have the same concerns regarding Slope-separated Terrace.

Response: Thank you for your insightful comment. For long-term, time-series land use classification, the 30-meter resolution Landsat imagery archive is the finest-resolution imagery freely and globally available. We acknowledge the inherent limitations posed by independently distributed terraces with surface areas smaller than 30 meters, making them challenging to identify on Landsat images. However, previous studies have shown that although individual terrace surfaces are smaller than 30 meters, they can be effectively detected when distributed in clusters (Cao et al., 2021; Li et al., 2024).

Regarding the question of whether Landsat imagery can effectively distinguish zig terraces or slopeseparated terrace, we conducted a systematic analysis using data from the China's First National Census for Water. We found that both terrace types are primarily used for cultivating cash crops and exhibit clustered spatial distributions. This concentrated spatial distribution creates distinct contrasts with surrounding land cover types (such as tea and rubber), thereby enabling their effective identification in Landsat imagery.


and some are in desert regions). Even without machine learning methods, it is entirely possible to determine this, as terraced fields simply cannot exist in these regions. This suggests that the subsequent accuracy validation may be inflated. The author should consider balancing the number of sample points across different types.

Response: Thank you for your insightful comment. In the revised manuscript, all sample points from manual visual interpretation based on high-resolution images from Google Earth Pro software were used to train the model, as shown in Figure S1. The statistical information of terrace and non-terrace samples for classification is presented in Table S6. To ensure that the model learns sufficient non-terrace characteristics, we also included sample points from desert regions and the Himalayan Mountain range during model training. For accuracy validation, we conducted a comprehensive validation using 14,986 high-quality field survey samples derived from China's First National Census for Water (Figure 2). None of validation samples were located in desert regions and the Himalayan Mountain range, thereby avoiding inflation of validation accuracy. Among the validation sample, the number of zig terrace and slope-separated terrace was relatively small. This aligns with the distribution characteristics of China's terraces. Analysis of data from the China's First National Census for Water reveals that, compared to level terraces and slope terraces, China has few zig terraces and slope-separated terraces.

Figure S1. The spatial distribution of train samples in 2020.

Table S6. Train samples collected from 2000 to 2010.

| Type          | 2000 | 2010 | 2020 |
|---------------|------|------|------|
| Level terrace | 9230 | 8532 | 8344 |
| Slope terrace | 6267 | 6271 | 6234 |
| Zig terrace   | 1159 | 1228 | 1248 |

| Slope-separated terrace | 301   | 415   | 414   |
|-------------------------|-------|-------|-------|
| Non-terrace             | 17934 | 17626 | 18171 |
| Total                   | 34891 | 34072 | 34411 |

Figure 2. Spatial distribution of validation samples.

Table S7. Validation samples in 2010.

| Non-
terrace | Level
terrace | Slope
terrace | Zig
terrace | Slope-separated terrace | Total |
|-----------------|------------------|------------------|----------------|-------------------------|-------|
| 11280           | 2998             | 584              | 104            | 20                      | 14986 |

5. The authors primarily focus on analyzing the spatiotemporal patterns within the dataset and evaluating variable importance, while paying comparatively less attention to methodological validation and uncertainty quantification. The authors lack a comprehensive assessment of the data quality. In addition, the authors did not provide visual results for the reference true values, making it difficult for readers to judge the validity of the results.

Response: Thank you for the valuable suggestion. In the revised manuscript, we restructured the paper and strengthened the analysis of data quality. We removed unnecessary sections (Section 3.3 in the original version), added new content in Sections 3.2 (Lines 217-231) and 4.1 (Lines 278-289), and revised Section 4.3 (Lines 314-332) and Section 4.4 (Lines 334-360) to strengthen the evaluation of data quality. Furthermore, we conducted a comprehensive validation using 14,986 high-quality field survey samples derived from China's First National Census for Water and added a spatial distribution map of the validation samples in Section 2.5 of the Methods (Figure 2).

Figure 2. Spatial distribution of validation samples.

Accordingly, the sentences (Lines 251-257) have been revised as follows:

Section 3.2 (Lines 217-231):

[revised manuscript text omitted]

6. Table 4 shows notably low F1-scores for Level terrace, Zig terrace and slope-separated terrace, suggesting limited model performance in discriminating these specific terrace types. The classification methodology requires optimization to achieve satisfactory accuracy levels that meet the scientific objectives.

Response: Thank you for your helpful comment. We improved the classification methodology and reprocessed the dataset to achieve a satisfactory accuracy level. Specifically, to alleviate the impact of crop spectral variability on classification accuracy, the study area was divided into six subregions (Figure 3). Within each region, different terrace types were classified separately. Given the sensitivity of the RF model to the ratio of samples across different classes, we implemented a two-stage mapping approach for classifying terraces within each region. The first stage focused on identifying terraces, while the second stage distinguished different terrace types. Additionally, we increased the number of sample points in different regions based on high-resolution images from Google Earth Pro software. In the

revised manuscript, the average overall accuracy for different terrace types reached 89.4%, with an average F1-score of 78.9%. Level terraces, zig terraces, and slope-separated terrace all achieved satisfactory classification accuracy (Table 4).

Table 4. The accuracy matrix for the different types of terraces.

| Year | types                                  | UA (%) | PA (%) | F1 score (%) | OA (%) | KA (%) |
|------|----------------------------------------|--------|--------|--------------|--------|--------|
| 2000 | Level terrace                          | 93.7   | 94.1   | 93.9         |        |        |
|      | Slope terrace                          | 70.1   | 70.6   | 70.3         | 00.7   | 66     |
|      | Zig terrace                            | 74.6   | 64.1   | 68.9         | 89.7   | 66     |
|      | Slope-separated terrace 85.7 70.6 77.4 |        |        |              |        |        |
| 2010 | Level terrace                          | 93.8   | 94     | 93.9         |        | 69.5   |
|      | Slope terrace                          | 73.1   | 73.2   | 73.2         | 89.8   |        |
| 2010 | Zig terrace                            | 77.6   | 68.6   | 72.8         | 09.0   |        |
|      | Slope-separated terrace                | 83.3   | 88.2   | 85.7         |        |        |
|      | Level terrace                          | 93.7   | 92.9   | 93.3         |        |        |
| 2020 | Slope terrace                          | 67.8   | 71.3   | 69.5         | 88.8   | 65.1   |
| 2020 | Zig terrace                            | 70     | 68.8   | 69.4         | 00.0   | 03.1   |
|      | Slope-separated terrace                | 86.7   | 72.2   | 78.8         |        |        |

**Figure 3.** Geographical regionalization in China. SW represents Southwest China. NW represents Northwest China. NENC represents Northeast and North China. SC represents South China. CC represents Central China. EC represents East China.

7. The authors compare their results with the 30-meter resolution terrace dataset published in ESSD (2021). However, as shown in Figure 3, the validation areas are geographically proximate. This limited spatial scope reduces the robustness of the comparative analysis. To more convincingly demonstrate the advantages of their approach, the authors should expand the comparison to include diverse geographic regions representing different environmental conditions and terrace types.

To ensure the robustness and geographical representativeness of comparative analysis, we expanded the spatial scope to include multiple geographic regions. These regions include diverse geographical environments and major terrace types across China, including North China, Northwest China, Southwest China, Central China, and East China. The figure 4 has been updated accordingly:

**Figure 4.** Regional comparisons of the three terraces datasets. (a) The distribution of cropland in China in 2020. (b) The spatial distributions of the three terraces datasets.

8. The phrase "from 2000 to 2020" in the author's title may mislead readers into assuming that the dataset is on an annual scale, but in fact the author only produced three years.

Response: Thank you very much for your valuable suggestion regarding the title. Based on all reviewers' comments on the title, we have revised the title to "A 30 m resolution dataset of soil and water conservation terraces across China for 2000, 2010, and 2020" (Lines 1-2).

9. After constructing feature factors, authors should perform correlation analysis between factors, which is a necessary step in feature construction. In addition, authors should also provide more detailed formulas or steps for calculating each factor.

Response: Thank you for your constructive comments. After constructing feature factors, to eliminate multicollinearity among the feature variables, we removed highly correlated features based on two criteria: (a) a variance inflation factor (VIF) value for each feature less than 10, and (b) pairwise Pearson correlation coefficients are below 0.7 (Liao et al., 2021). Detailed information about the used features is provided in Table S3, Table S4, and Table S5 of the supplementary materials. Table S5 is shown below:

Table S5. Used features of multi-temporal metrics in 2020 for SWCTMD mapping.

| Region                          | Features for classifying terrace                                                                                                                                               | Features for classifying various terrace types                                                                                                                |
|---------------------------------|--------------------------------------------------------------------------------------------------------------------------------------------------------------------------------|---------------------------------------------------------------------------------------------------------------------------------------------------------------|
| Southwest
China              | SOS, SR_B2_p25, SR_B5_p25, SR_B5_p75, SR_B7_p75, aspect, bsi_p75, elevation, evi_p25, evi_p75, lswi_p25, lswi_p75, mndwi_p25, mndwi_p75, P, slope                              | SOS, SR_B2_p75, SR_B5_p25,
SR_B5_p75, aspect, elevation,
lswi_p25, mndwi_p25,
ndvi_p75, P, slope                                                     |
| Northwest
China              | SOS, SR_B2_p50, SR_B5_p25,
SR_B5_p75, SR_B7_p25, aspect,
bsi_p50, bsi_p75, elevation, evi_p25,
evi_p50, evi_p75, lswi_p25, mndwi_p25,
ndvi_p25, ndvi_p75, P, slope | SOS, SR_B2_p25, SR_B2_p75,
SR_B5_p50, SR_B5_p75,
aspect, elevation, evi_p25,
lswi_p25, lswi_p75,
mndwi_p25, mndwi_p75, P,
slope                |
| Northeast
and North
China | SOS, SR_B2_p50, SR_B5_p25,
SR_B5_p75, SR_B7_p25, aspect,
bsi_p50, bsi_p75, elevation, evi_p25,
evi_p50, evi_p75, mndwi_p25,
ndvi_p25, P, slope                     | SOS, SR_B2_p25, SR_B5_p75,
aspect, bsi_p25, elevation,
evi_p25, evi_p50, evi_p75,
lswi_p25, lswi_p50,
mndwi_p25, mndwi_p75,
ndvi_p25, P, slope |
| South
China                  | SOS, SR_B2_p50, SR_B2_p75,
SR_B5_p25, SR_B5_p75, SR_B6_p75,
aspect, elevation, lswi_p25, lswi_p75,
mndwi_p25, P, slope                                                | SOS, SR_B2_p25, SR_B2_p50,
SR_B2_p75, SR_B5_p25,
SR_B5_p75, SR_B7_p25,
aspect, elevation, lswi_p25,
mndwi_p25, mndwi_p75,
ndvi_p75, P, slope   |
| Central
China                | SOS, SR_B2_p75, SR_B3_p25,
SR_B5_p50, SR_B6_p75, aspect,
elevation, evi_p25, lswi_p25, lswi_p75,
mndwi_p75, ndvi_p75, P, slope                                        | SOS, SR_B2_p25, SR_B2_p75,
SR_B5_p25, SR_B5_p75,
aspect, elevation, lswi_p75,
mndwi_p25, ndvi_p25, P, slope                                          |
| East China                      | SOS, SR_B2_p75, SR_B5_p25, SR_B5_p75, SR_B6_p75, aspect, elevation, evi_p25, lswi_p25, lswi_p75, mndwi_p75, P, slope                                                           | SOS, SR_B2_p75, SR_B5_p25, SR_B5_p75, aspect, elevation, lswi_p25, lswi_p75, mndwi_p25, mndwi_p75, P, slope                                                   |

The authors mention that ignoring terrace types could lead to inaccuracies in soil erosion assessments. I recommend expanding on this point to clarify the mechanisms through which different terrace types influence erosion rates. This would help to better justify the necessity of developing a terrace-type-specific dataset.

Response: Thank you for your insightful comment. In the Introduction of the revised manuscript (Lines 61-68), we further elaborated on the mechanisms by which different terrace types influence soil erosion rates. It now stands as follows:

"The effectiveness of terraces in SWC varies according to type. Level terraces, characterized by flat cultivated surfaces, can effectively reduce the amount, velocity, and energy of surface runoff and increase water infiltration, thereby effectively preventing the transportation of sediment (Wei et al., 2012; Chen et al., 2013; Arnáez et al., 2015). Zig terraces increase water infiltration and reduce runoff by creating micro-catchments (Wang et al., 2004). Conversely, slope terraces, with their uneven surfaces, are more prone to generating runoff than level terraces or zig terraces (Wei et al., 2016). Level terraces exhibit the most effective SWC benefits (Oliveira et al., 2012). Compared to slope terraces, level terraces can reduce runoff by 56.5% and sediment by 53.1% (Chen et al., 2017). Ignoring terrace type can lead to inaccuracies in soil erosion assessment, and the absence of long-term terrace data hinders analyses of soil erosion trends."

**2. Spatial Resolution Clarification:**

Please explicitly state the spatial resolution of the final product in the Data and Methods section. Additionally, if input datasets used in the production process have differing spatial resolutions, please clarify how this issue was handled (e.g., resampling methods or resolution harmonization techniques).

Response: Thank you for your advice. We have explicitly specified the spatial resolution of the final product in the Methods section (Line 82-84). In this study, we used only 30 m resolution Landsat imagery and 30 m resolution Copernicus DEM data, without performing any data resampling or resolution conversion. In addition, we provided detailed information on the data in the supplementary materials (Table S1).

"Figure 1 illustrates the framework of 30-meter resolution terrace mapping. The workflow includes sample collection, feature calculation, classification implementation, post-classification processing, and accuracy evaluation. Detailed information on each stage of the terrace mapping process is provided below."

Table S1. The multitemporal data series used in this study.

| Data name                                          | Year             | Spatial resolution (m) | Data sources                                                                                                                    |
|----------------------------------------------------|------------------|------------------------|---------------------------------------------------------------------------------------------------------------------------------|
| Landsat-5/8
surface
reflectance
(SR) data | 2000, 2010, 2020 | 30                     | The data is accessible via GEE and is provided by the United States Geological Survey (USGS) (https://earthengine.google.com/). |
| Copernicus
DEM                                  | 2010             | 30                     | The data is accessible via GEE (https://earthengine.google.com/).                                                               |
| GlobeLand30                                        | 2000, 2010, 2020 | 30                     | The data is provided by the National Geomatics Center of China (NGCC) (http://www.globallandcover.com/).                        |

**3. Validation with Statistical Data:**

Has the dataset been validated or cross-referenced with any official statistical records on terrace areas, either at the national or regional level? If feasible, a comparison with statistical data, including breakdowns by terrace type, would strengthen the credibility and applicability of the dataset.

Response: Thank you for the valuable suggestion. Due to the lack of official terrace area statistics at the national and regional levels, we are unable to conduct comparative verification. To ensure the reliability of the dataset, we evaluated dataset accuracy using 14,986 field survey samples from the China's First National Census for Water. The results indicate that the dataset exhibits high accuracy. The average overall accuracy (OA) of the terrace was 91.7% and the average F1 score was 83.4%. For different terrace types, the average OA was 89.4% and the average F1 score was 78.9%.

**4. Comparison with Existing Datasets:**

To further highlight the value and potential advantages of the newly developed dataset, I suggest comparing it with existing terrace-related datasets (if available). This could include spatial consistency, classification accuracy, or coverage of different terrace types.

Response: Thank you for your constructive comments. At the national scale, only two terrace datasets are currently available: the 2018 China Terrace Map (CTM2018) (Cao et al., 2021) and the 2017 China Terrace Map (CTM2017) (Li et al., 2024). These datasets do not further classify terrace types, so we were only able to compare terrace coverage. In the revised manuscript, we conducted a comparative analysis of product consistency across different geographic regions in China (Figure 4). Additionally, due to the lack of validation samples for 2017 and 2018, we were unable to assess the classification

accuracy of CTM2018 and CTM2017. To analyze the differences among the three datasets, In Section 4.1, we compared terrace areas at the provincial scale and analyzed the causes of these discrepancies. It now stands as follows (Lines 203-212, and 278-288):

"Figure 4 illustrates the spatial consistency between SWCTMD and the two existing datasets: the 2018 China Terrace Map (CTM2018) (Cao et al., 2021) and the 2017 China Terrace Map (CTM 2017) (Li et al., 2024). SWCTMD exhibits the highest accuracy. Compared with SWCTMD and CTM2018, CTM2017 exhibits relatively lower accuracy for both typical terrace and non-terraces areas (regions A, B, C, D F and G in Fig. 4b). For typical terraces, SWCTMD and CTM2018 show similar identification performance (regions A, B, C and F in Fig. 4b). However, for atypical terraces, such as zig terraces located in Yunnan Province, SWCTM successfully identifies these as terraces, whereas CTM2018 fails to identify them as terraces (regions E in Fig. 4b). Conversely, for non-terrace areas situated in the Middle-Lower Yangtze River, SWCTMD accurately classifies these as non-terraces, while CTM2018 erroneously identify them as terrace areas (regions G in Fig. 4b). At the provincial scale, the majority of provinces exhibit larger terrace areas in SWCTMD compared to both CTM2018 and CTM2017 (Tables S8 and S9)."

"We compared the 2020 terrace area estimated by SWCTMD with those from CTM2018 and CTM2017. SWCTMD exhibited the largest terrace area compared to CTM2018 and CTM2017. The areal discrepancies can be attributed to the following reasons. First, CTM2017 and CTM2018 predominantly focused on the most typical level terraces, whereas our research encompasses a broader range of terrace types, including non-typical terraces such as slope terraces, zig terraces, and slope-separated terrace. Second, each dataset employed distinct cropland for terrace classification. SWCTMD utilized the union of cropland with slopes exceeding 2° from the 2000, 2010, and 2020 GlobeLand30 cropland data, whereas CTM2018 employed only the 2010 GlobeLand30 cropland data, and CTM2017 adopted FROM-GLC cropland data. Third, CTM2018 excluded isolated patches smaller than 9,000 m² from its classification scheme. However, since SWCTMD constrains its classification to cropland with slopes exceeding 2°, the identified terrace areas in Anhui, Fujian, Jiangxi, and Zhejiang provinces were smaller than those from CTM2018. In these provinces, CTM2018 included terraces with slopes below 2°, which is classified as non-terraces according to the technical regulations of the third nationwide land survey. Overall, our dataset provides more comprehensive coverage for terraces and exhibits higher accuracy and robustness."

Figure 4. Regional comparisons of the three terraces datasets. (a) The distribution of cropland in China in 2020. (b) The spatial distribution of the three terraces datasets.